# STRUCTURAL MULTI-VIEW CLUSTERING NETWORK VIA HETEROGENEOUS RANDOM WALKS

## ABSTRACT

Multi-view clustering, which aims to partition data samples into disjoint clusters by leveraging information from multiple views, has been shown to be highly effective when incorporating structure information, as widely acknowledged in recent works. This paper presents a novel structural multi-view clustering network via heterogeneous random walks, guided by a unified sample-level structure to enhance clustering performance. We first construct a multi-view heterogeneous graph consisting of sample nodes and view nodes, capturing correlations between views while preserving their specific structures. Then, a multi-step random walk strategy on the heterogeneous graph is introduced to explore high-order sample structures across various views, ensuring that each view structure is taken into account. Based on this, a lightweight network is designed to facilitate structure learning both within-view and cross-view, guided by the unified structure derived from heterogeneous random walks, ultimately achieving representations that are conducive to clustering. Extensive experiments on five real-world datasets demonstrate the superiority of the proposed method.

## 1 INTRODUCTION

Multi-view clustering (MVC), which exploits the multi-view information to partition data samples into disjoint clusters, has garnered significant attention in recent years. For example, DEMVC (Xu et al., 2021) explores deep embedded multi-view clustering with collaborative training, learning individual view representations and refining them collaboratively to enhance clustering performance. SCMRL (Zhou et al., 2023) seeks to uncover underlying multi-view semantic consensus information and leverage it to guide the learning of integrated feature representations.

Despite the success of these methods, they primarily focus on data representation learning and often neglect the structure information of the data. However, the importance of accounting for structures among data samples has been extensively recognized, as they reveal hidden similarities and associations among samples that extend beyond representation spaces, particularly in the context of multi-view data. Such structure is valuable for representation learning, further facilitating the discovery of latent patterns within the data. Recently, some methods have attempted to utilize structure information to explore the intrinsic relationships among data samples. GDMVC (Bai et al., 2024b) combines both feature and structural information using a graph convolutional autoencoder to more effectively capture the underlying data relationships. DIVIDE (Lu et al., 2024) introduces high-order random walks within each individual view to learn latent structure information, which is then used to guide multi-view representation learning and promote subsequent clustering.

Existing methods learn representations by incorporating view-specific structure information, but most of them still have a gap in discovering a unified sample-level data partition. Notably, in multi-view scenarios, data samples are described by different views, meaning that such data is characterized by multiple view representations rather than explicit sample representations. That is to say, in multi-view clustering tasks, we can only utilize the view structure to assist in obtaining sample partitions, instead of using the sample structure. Existing methods such as (Bai et al., 2024b; Lu et al., 2024) have made considerable attempts to utilize view structures, but they share common limitations: obvious differences between the structures of multiple views make it difficult to achieve a consistent sample structure across views, potentially leading to the collapse of the representations; additionally, each view exhibits its own specific structure, with no evidence to support one another,

making it challenging to determine which view's structure should contribute more to the sample structure, to say nothing of guaranteeing the optimality of the sample structure. Therefore, the structure information required for multi-view clustering tasks should have the following attributes: on one hand, it should retain view-specific structure information to reflect the characteristics of the data from different views, even if these structures may differ; on the other hand, it should extend to the sample level to provide a comprehensive and unified guidance for effective clustering.

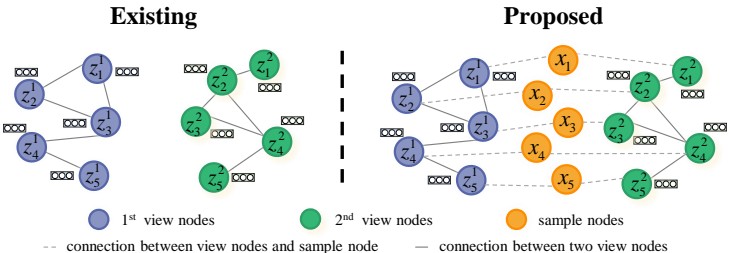

Figure 1: The existing view graph and the proposed heterogeneous graph of two-view data. Clearly, we preserve the view structures while introducing sample nodes that lack explicit representations to establish the correlation between different view graphs.

In this paper, we propose a novel structural multi-view clustering network via heterogeneous random walks, guided by a unified sample-level structure through hybrid structure learning, termed SMvCNet. To obtain the structure information required for multi-view clustering, we construct a multi-view heterogeneous graph (shown in Figure 1), which contains two types of nodes: sample nodes and view nodes. In detail, sample nodes are connected through their affiliation with view nodes, while view nodes retain the view structure information. By constructing such a graph, we can better capture the correlations between views while preserving their distinct structures, thus providing a foundation for multi-view learning. Then, we design a multi-step random walk strategy on the constructed heterogeneous graph. By leveraging the designed random walks, we uncover shared high-order sample structures across different views while taking view structures into account; during these walks, the contributions of the view structures are automatically integrated, allowing us to select the optimal sample structure. Finally, a lightweight network is introduced that jointly performs hybrid structure learning, guided by the unified sample-level structure derived from heterogeneous random walks. This learning process incorporates both within-view and cross-view structure to produce clustering-friendly representations and can be trained end-to-end for clustering tasks. Extensive experiments conducted on five real-world datasets demonstrate the superiority of the proposed network over existing multi-view clustering models.

## 2 RELATED WORK

Multi-view clustering(Bickel & Scheffer, 2004; Fang et al., 2023) aims to achieve enhanced clustering performance by integrating information from different views. With the growing diversity of data forms, multi-view clustering has emerged as one of the most popular tasks in data mining. Based on the design of multi-view learning objectives, this paper classifies existing multi-view clustering methods into two categories: traditional representation-based multi-view clustering and structural multi-view clustering methods that take into account relationships among data samples. With the rise of deep learning, representation-based deep multi-view clustering has seen substantial advancements. DEMVC (Xu et al., 2021) is a deep embedded multi-view clustering model with collaborative training, where individual view representations are learned and then refined collaboratively to enhance clustering performance. Yan et al. (2023) presented a multi-view clustering framework that aggregates global and cross-view features to learn a consensus representation for enhancing clustering. MFLVC (Xu et al., 2022) is designed through multi-level feature learning for contrastive multi-view clustering to improve clustering performance. CVCL (Chen et al., 2023) learns invariant representations by contrasting cluster assignments across views. Despite these advancements, challenges such as incomplete multi-view data persist. To tackle these, recent studies have incorporated contrastive learning and self-supervised objectives to learn more discriminative representations (Yang et al., 2022; Li et al., 2023; Chao et al., 2024). The core of clustering tasks lies in the similar-

ity among samples. As a result, many researchers argue that incorporating the structural information of samples in multi-view representation learning can improve clustering performance(Wang et al., 2023; Yang et al., 2023; Bai et al., 2024a). Liu et al. (2022) used neural networks to estimate stationary diffusion states, leveraging graph neural networks for multi-view learning. Gu et al. (2024) introduced a multi-view clustering method (ONESELF), which integrates robust latent representation extraction with target structured graph construction into a cohesive optimization framework. DIVIDE(Lu et al., 2024) is a robust contrastive multi-view clustering method that uses high-order random walks to consider sample relationships. SURER (Wang et al., 2024a) refines view-specific attribute graphs and integrates them into a unified heterogeneous graph, which is used by a graph neural network to learn a consensus representation for clustering. In addition, there are also some approaches that introduce structural information through subspace learning. For example, Si et al. (2022) proposed to enhance multi-view clustering by enforcing diversity among views and applying a structure constraint to the subspace self-representation. And Wang et al. (2024b) developed a framework that jointly learns a latent subspace and structured graph for multi-view clustering.

## 3 METHOD

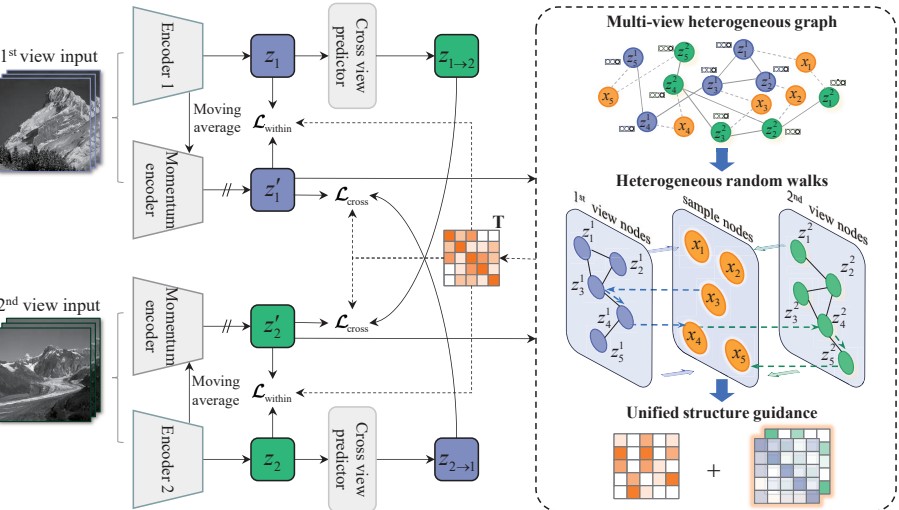

Figure 2: The proposed structural multi-view clustering network via heterogeneous random walks.

In this paper, we propose a structural multi-view clustering network via heterogeneous random walks, and the detailed fowchart is illustrated in Figure 2. We first introduce the construction of multi-view heterogeneous graph and how to use it acquire unified sample structure guidance across multiple views. Furthermore, the unified structure guidance serves as a target for hybrid structure learning from both within-view and cross-view to enhance multi-view clustering.

### 3.1 MULTI-VIEW RANDOM WALKS ON HETEROGENEOUS GRAPH

To explore the unified sample structure across multiple views, we proposed employing multi-view multi-step random walks to progressively identify the high-order neighbors of each data sample. With the high-order neighbors across all views and the heterogeneous affinity, our method could model the consistency and complementariness of multi-view data. For ease of presentation, we briefly introduce the preliminary on traditional random walk. Specifically, a random walk on a graph is a process that starts at some nodes and moves to other nodes according to the edge weights at each step (Lovász, 1993). In other words, it can be represented by as a transition matrix $\mathbf{M}$. And each element $\mathbf{M}_{ij}$ indicates the probability of moving from the $i$-th node to the $j$-th node in one step. In this way, the probability after multi-step random walks could be denoted by:

$$p(t) = p(t-1)\mathbf{M} = \cdots = p(0)\mathbf{M}^t \tag{1}$$

where $t$ is the transmission step.

Different from traditional random walks, constructing the unique transition matrix of each view and using it to mine the consistency and complementarity within multi-view data is an urgent problem that needs to be solved. Therefore, we use each data sample as starting node and perform $t$ step multi-view random walks to find the high-order neighbors. More specifically, to build the sample relationship within each specific view, we first construct a fully-connected adjacency matrix with the in-batch samples for each view, by regarding the embedding as nodes and defining the edge weights with the following Gaussian kernel similarity:

$$\mathbf{A}_{ij}^v = \exp(-\frac{1 - (\mathbf{h}_i \cdot \mathbf{h}_j^T)}{\gamma}) \tag{2}$$

where $\mathbf{h}_i$ is the $i$-th embedding in $v$-th view, and $\gamma$ is bandwidth parameter fixed as $0.1$. In this way, the transition matrix $\mathbf{M}^v$ of $v$-th view is obtained by normalizing the adjacency matrix $\mathbf{A}^v$ in a row-wise manner. For achieving robustness across multiple views, we exploit all transition matrices $\{\mathbf{M}^v\}_{v=1}^V$ to perform the designed heterogeneous random walks.

**Heterogeneous Graph construction** To integrate structure information from different views without disrupting the view-specific structures, we construct a heterogeneous graph containing two types of nodes, sample node and view node. Specifically, the sample node has no explicit representation, it can only be linked to the view node; and the view node is represented by the hidden embedding, which can be linked to both the view node and the sample node. For each sample node, it is linked to $V$ view nodes describing the sample with same probability, and there are no links between sample nodes. For each view node, it is linked to other view nodes according its corresponding transition probability; and each view node must be linked to its sample node. According to this, we can explore structural relationships among multi-view sample nodes through the weights between view nodes.

**Heterogeneous Multi-step Random Walks** Consider a dataset with $N$ samples, represented by the set of **sample nodes** $X = \{x_1, x_2, \ldots, x_N\}$. Each sample node $x_i$ is associated with $V$ different views, represented by the set of **view nodes** $z_i = \{z_i^1, z_i^2, \ldots, z_i^V\}$, where $z_i^v$ represents the $v$-th view node of sample node $x_i$. The view nodes relationship in $v$-th view is represented by $\mathbf{M}^v$. Our proposed multi-step random walk on heterogeneous graph can be described as following processes:

(1) Multi-view Single-Step Random Walk Process

The multi-view random walk consists of three main stages in a single step:

- i. Initial Transition from Sample Node to View Node:

Start from a sample node $x_i$ and randomly choose a view index $\nu \in \{1, 2, \ldots, V\}$ corresponding to a view node $z_i^\nu$. The probability of selecting each view is uniformly distributed:

$$P(z_i^\nu \mid x_i) = \frac{1}{V}, \quad \text{where } \nu \in \{1, 2, \ldots, V\}$$

- ii. Transition between View Nodes:

For each sample $x_i$, given the selected view $\nu$, transition from the view node $z_i^\nu$ to another view node $z_k^\nu$ within the same view. The transition probability is defined by the transition matrix $\mathbf{M}^\nu$ of view $\nu$:

$$P(z_k^\nu \mid z_i^\nu, \mathbf{M}^\nu) = \mathbf{M}_{ik}^\nu$$

- iii. Return Transition from View Node to Sample Node:

After reaching the view node $z_k^\nu$, transition back to the corresponding sample node $x_k$ with probability 1:

$$P(x_k \mid z_k^\nu) = 1$$

Thus, the transition probability from sample node $x_i$ to sample node $x_k$ for a single-step multi-view random walk is:

$$P(x_k \mid x_i) = \sum_{\nu=1}^{V} \left( P(z_i^\nu \mid x_i) \cdot P(z_k^\nu \mid z_i^\nu, \mathbf{M}^\nu) \cdot P(x_k \mid z_k^\nu) \right)$$

$$= \frac{1}{V} \sum_{\nu=1}^{V} \mathbf{M}_{ik}^\nu \tag{3}$$

(2) Multi-view Multi-Step Random Walk Process

The multi-step random walk can be seen as an iteration of the above single-step process. At each step $l$, a view $\nu(l)$ is randomly chosen, and the above single-step transition is performed.

For the multi-step transition process, considering the independence and randomness of view selection at each step, the $t$-step transition probability matrix from sample node $s_i$ to sample node $s_k$, denoted as $\mathbf{M}^{(t)}$, can be expressed as:

$$\mathbf{M}^{(t)} = \mathbb{E}\left[\prod_{l=1}^{t} \mathbf{M}^{\nu(l)}\right]$$

where $\nu(l) \in \{1, 2, \ldots, V\}$ represents the set of view selections for all sample nodes at step $l$. where the expectation $\mathbb{E}$ is taken over all possible view selection sequences $(\nu(1), \nu(2), \ldots, \nu(t))$ with a uniform distribution.

Therefore, the $t$-step transition probability matrix $\mathbf{M}^{(t)}$ can be represented as:

$$\mathbf{M}^{(t)} = \frac{1}{V^t} \sum_{\nu(1)=1}^{V} \sum_{\nu(2)=1}^{V} \cdots \sum_{\nu(t)=1}^{V} \left(\prod_{l=1}^{t} \mathbf{M}^{\nu(l)}\right). \tag{4}$$

Through this algorithm, we achieve a full walking paths heterogeneous graph that integrates the structure information from multiple views, capturing complex relationships and enhancing the representational power. The resulting transition matrix $\mathbf{M}^{(t)}$ thus provides a robust unified structure guidance at sample level for multi-view clustering.

Considering the complexity of calculating $\mathbf{M}^{(t)}$, we introduce a hyper-parameter $n_{\mathrm{rw}}$ to control the number of rounds in the heterogeneous random walks. This allows us to obtain high-confidence walk paths while avoiding the computation of all paths. The resulting matrix is denoted as $\tilde{\mathbf{M}}^{(t)}$.

## 3.2 Unified Structure Learning for Multi-view Clustering

In this part, we develop a structural multi-view clustering network via heterogeneous random walks, guided by a unified sample-level structure through hybrid structure learning from both within-view and cross-view. The proposed network is composed of a series of siamese within view encoders and cross view predictors. Each siamese within view encoder is employed to learn semantic representation of each view. The cross view predictor projects one view representation to others' view representations by applying two-layer non-linear fully connected networks. Specifically, the overall objective function of proposed SMvCNet is formulated as follows:

$$\mathcal{L} = \mathcal{L}_{\mathrm{within}} + \mathcal{L}_{\mathrm{cross}} \tag{5}$$

where $\mathcal{L}_{\mathrm{within}}$ and $\mathcal{L}_{\mathrm{cross}}$ are the within-view structure learning loss and cross-view structure learning loss, respectively.

Let $(\psi_e^v(\cdot); \psi_{e'}^v(\cdot))$ be the parameters of the siamese within view encoders for the $v$-th view. The within-view structure learning loss $\mathcal{L}_{\mathrm{within}}$ is formulated as follows:

$$\mathcal{L}_{\mathrm{within}} = \sum_{v=1}^{V} \mathcal{J}(\mathbf{T}, \sigma(\psi_e^v(\mathbf{X}^v), \psi_{e'}^v(\mathbf{X}^v))) \tag{6}$$

where $\mathcal{J}(p, q)$ indicates the cross entropy, $\mathbf{T} \in \mathbb{R}^{n \times n}$ is the pseudo target that contains the sample relationships across multiple views, and $\sigma(\mathbf{A}, \mathbf{B})$ is the pairwise similarity $\mathrm{sim}(\cdot, \cdot)$ with the row-wise normalization operator, i.e.,

$$[\sigma(\mathbf{A}, \mathbf{B})]_{ij} = \frac{\exp(\mathrm{sim}(\mathbf{A}_i, \mathbf{B}_j)/\tau)}{\sum_{r=1}^{n} \exp(\mathrm{sim}(\mathbf{A}_i, \mathbf{B}_r)/\tau)} \tag{7}$$

where $\tau$ is the temperature fixed to 0.5 throughout our experiments.

Let $\psi_p^v(\cdot)$ be the parameters of the cross view predictor for the $v$-th view. The cross-view structure learning loss $\mathcal{L}_{\text{cross}}$ is formulated as follows:

$$\mathcal{L}_{\text{cross}} = \sum_{v=1}^{V} \sum_{u \neq v}^{V} \mathcal{J}(\mathbf{T}, \sigma(\psi_p^u(\psi_e^u(\mathbf{X}^u)), \psi_{e'}^v(\mathbf{X}^v))) \tag{8}$$

To achieve unified structure learning across multiple views, a pseudo target in equation 6 and equation 8 is built as unified structure guidance by integrating multi-view first-order and high-order ($t$-order) structure information, i.e.,

$$\mathbf{T} = \alpha\tilde{\mathbf{M}} + (1 - \alpha)\tilde{\mathbf{M}}^{(t)} \tag{9}$$

where $\alpha$ is a balanced parameter fixed to $0.5$. $\tilde{\mathbf{M}}$ indicates the multi-view $k$-nn matrix, which is defined as:

$$\tilde{\mathbf{M}} := g_{nor}(\sum_v \tilde{\mathbf{A}}^v) = g_{nor}(\sum_v g_k(\mathbf{A}^v)) \tag{10}$$

where $g_k(\cdot)$ is a matrix filter that determines the $k$ nearest neighbors of each sample and retains their corresponding weights. Specifically, each element $\tilde{a}_{ij}^v \in \tilde{\mathbf{A}}^v$ is set to $a_{ij}^v \in \mathbf{A}^v$ if the $i$-th sample and $j$-th sample are regarded as neighbor pair in the $v$-th view; and $\tilde{a}_{ij}^v$ is set to 0 otherwise. And $g_{nor}(\cdot)$ indicates the matrix normalization in a row-wise manner.

Overall, for within-view structure learning, $\mathbf{T}$ is utilized to improve within-view discrimination while incorporating enhanced information from other views. For cross-view structure learning, $\mathbf{T}$ is used to enhance cross-view interactions, facilitating the exploration of the consistent and complementary information across multiple views.

## 4 EXPERIMENTS

### 4.1 DATASET

To evaluate the effectiveness of SMvCNet, we carry out extensive experiments across five benchmark datasets: Scene15, UCI-Digit, CUB, Prokaryotic and ALOI. A succinct overview of these datasets is presented in Table 1. In this section, we implement experiments to verify the effectiveness of the proposed SMvCNet through answering the following questions: (**Q1**) Does SMvCNet outperform state-of-the-art multi-view clustering methods? (**Q2**) Does each component of SMvCNet contribute to the overall performance? (**Q3**) How do the hyper-parameters impact the performance of SMvCNet? (**Q4**) What is the clustering structure revealed by SMvCNet? (**Q5**) How to reduce the complexity of multi-view heterogeneous graph random walks?

Table 1: Summary of the utilized benchmark datasets.

| Dataset | Samples | Views | Clusters | Dimensions |
|---|---|---|---|---|
| Scene15(Yang et al., 2022) | 4,485 | 2 | 15 | 20/59 |
| UCI-Digit(Bai et al., 2024a) | 2,000 | 3 | 10 | 76/216/64 |
| CUB(Li et al., 2023) | 600 | 2 | 10 | 1024/300 |
| Prokaryotic(Brbić et al., 2016) | 551 | 3 | 4 | 393/3/438 |
| ALOI(Cui et al., 2024) | 10,800 | 4 | 100 | 77/13/64/125 |

### 4.2 EXPERIMENT SETUP

The experiments are conducted utilizing the following hardware setup: Intel Xeon Platinum 8370C CPU and NVIDIA RTX A6000 GPU. Additionally, the PyTorch platform is employed for all experiments. In the case of SMvCNet, we utilize the Adam (Kingma & Ba, 2015) optimizer to minimize the total loss equation 5.

**Comparison Methods** The proposed SMvCNet is benchmarked against ten prominent deep multi-view clustering algorithms. Specifically, these compared clustering algorithms can be broadly categorized into two groups: multi-view clustering models based on representation (Completer, ProImp, CVCL, ACCMVC and GCFAGG) and multi-view clustering models based on structure (SDSNE, SURER, ICMVC, MAGA and DIVIDE).

**Implementation Details** In the case of our SMvCNet, a batch size of 1024 is employed consistently across all datasets. The learning rate is set at 0.0005. We utilize the 4-layer fully connected network (FCN) and ReLU activation is employed to build the within-view encoder, and the cross-view predictor is a 2-layer FCN. To warm up, we set the target $\mathbf{T}$ in equation 9 as an identity matrix $\mathbf{I}$ for the first 100 epochs and then adopt the unified structure guidance ($k$=5) in the remaining epochs. In the experiments, we conducted 5 runs of the proposed SMvCNet and reported the average of the experimental results in the main experiments. For a comprehensive analysis, we use three widely used clustering metrics to evaluate the performance, including clustering Accuracy (ACC), Normalized Mutual Information (NMI) and Adjusted Rand Index (ARI). A higher value of these metrics indicates a better clustering performance.

## 4.3 COMPARISONS WITH STATE OF THE ARTS (Q1)

To demonstrate the superiority of the proposed SMvCNet, we compare SMvCNet with ten baselines, including deep multi-view clustering models based on representation and deep multi-view clustering models based on structure.

Table 2: Clustering performance across five multi-view benchmark datasets. The most exceptional results are marked in **bold**, and the second-best values are underlined. The "×" means the model can only apply on two views datasets. The "-" means we cannot get clustering results on the dataset because the large-scale dataset occupies many computation resources.

| Methods | Scene15 | | | UCI-Digit | | | CUB | | | Prokaryotic | | | ALOI | | |
|---|---|---|---|---|---|---|---|---|---|---|---|---|---|---|---|
| | ACC | NMI | ARI | ACC | NMI | ARI | ACC | NMI | ARI | ACC | NMI | ARI | ACC | NMI | ARI |
| Completer(Lin et al., 2021) | 0.3970 | 0.4225 | 0.2348 | × | × | × | 0.5850 | 0.7160 | 0.5389 | × | × | × | × | × | × |
| ProImp(Li et al., 2023) | 0.4499 | 0.4477 | 0.2738 | × | × | × | 0.8244 | 0.7653 | 0.6749 | × | × | × | × | × | × |
| CVCL(Chen et al., 2023) | 0.4216 | 0.4338 | 0.2523 | 0.9265 | 0.8728 | 0.8501 | 0.8019 | 0.7180 | 0.6291 | 0.5240 | 0.2773 | 0.2098 | 0.6834 | 0.8265 | 0.5990 |
| ACCMVC(Yan et al., 2024) | 0.4357 | 0.4224 | 0.2529 | 0.9030 | 0.8321 | 0.8016 | 0.7833 | 0.7049 | 0.6061 | **0.5844** | 0.3332 | 0.2242 | 0.8782 | 0.9338 | 0.8332 |
| SDSNE(Liu et al., 2022) | 0.4252 | 0.4391 | 0.2358 | 0.8170 | 0.8624 | 0.7816 | 0.7467 | 0.7627 | 0.6249 | 0.4628 | 0.2554 | 0.0872 | - | - | - |
| SURER(Wang et al., 2024a) | 0.4060 | 0.3868 | 0.2013 | 0.9010 | 0.8371 | 0.7897 | 0.8300 | 0.7647 | 0.6670 | 0.5064 | 0.2774 | 0.1069 | - | - | - |
| ICMVC(Chao et al., 2024) | 0.4276 | 0.4193 | 0.2487 | × | × | × | 0.7633 | 0.7357 | 0.6339 | × | × | × | × | × | × |
| MAGA(Bian et al., 2024) | 0.3561 | 0.3973 | 0.2076 | 0.9420 | 0.8811 | 0.8759 | 0.6700 | 0.7087 | 0.5643 | 0.5372 | 0.3103 | 0.1731 | 0.7356 | 0.8313 | 0.6307 |
| GCFAGG(Yan et al., 2023) | 0.3554 | 0.3813 | 0.2155 | 0.9120 | 0.8525 | 0.8204 | 0.7967 | 0.7493 | 0.6444 | 0.5154 | 0.3035 | 0.1948 | 0.8680 | 0.9351 | 0.8311 |
| DIVIDE(Lu et al., 2024) | 0.4305 | 0.4354 | 0.2587 | 0.9464 | 0.8972 | 0.8852 | 0.7537 | 0.7489 | 0.6483 | 0.5187 | 0.3163 | 0.1954 | 0.8418 | 0.8916 | 0.7522 |
| Ours | **0.4808** | **0.4887** | **0.3109** | **0.9538** | **0.9102** | **0.9007** | **0.8317** | **0.7891** | **0.7040** | 0.5539 | **0.3747** | **0.2330** | **0.9042** | **0.9449** | **0.8651** |

In Table 2, the clustering performance of all the compared methods on the five datasets is presented. Based on the results, we have the following observations: 1) In comparison to deep multi-view clustering methods, our SMvCNet consistently achieves the most favorable clustering outcomes across the majority of datasets. This indicates that building a heterogeneous graph based on multi-view data and employing the proposed multi-step random walks to obtain a unified structure guidance across multiple views is highly effective for multi-view representation learning and clustering tasks. 2) Thanks to the design of multi-view unified structure learning loss within and cross views, SMvC-Net explores the consistency and complementarity in multi-view data both within each view and between different views. 3) Compared to representation-based multi-view clustering methods, our SMvCNet is guided by unified structure, enabling the exploration of relationships between samples. This makes the learned representations more favorable for downstream tasks like clustering. 4) In contrast to existing multi-view structural clustering methods, such as DIVIDE, the proposed SMvCNet uses a heterogeneous random walk across all views to capture globally unified structure guidance, replacing the independent structure learning of each view. This can more effectively leverage the complementarity among various views, thereby improving clustering performance.

## 4.4 ABLATION STUDIES (Q2)

Table 3: Ablation studies on multi-view unified structure guidance.

| Method | Scene15 | | | UCI-Digit | | | CUB | | | Prokaryotic | | | ALOI | | |
|---|---|---|---|---|---|---|---|---|---|---|---|---|---|---|---|
| | ACC | NMI | ARI | ACC | NMI | ARI | ACC | NMI | ARI | ACC | NMI | ARI | ACC | NMI | ARI |
| None | 0.4544 | 0.4493 | 0.2794 | 0.9390 | 0.8837 | 0.8705 | 0.7500 | 0.7436 | 0.6399 | 0.4857 | 0.2839 | 0.1865 | 0.8254 | 0.8791 | 0.7279 |
| View-specific (DIVIDE) | 0.4305 | 0.4354 | 0.2587 | 0.9464 | 0.8972 | 0.8852 | 0.7537 | 0.7489 | 0.6483 | 0.5187 | 0.3163 | 0.1954 | 0.8418 | 0.8916 | 0.7522 |
| Unified (Ours) | **0.4808** | **0.4887** | **0.3109** | **0.9538** | **0.9102** | **0.9007** | **0.8317** | **0.7891** | **0.7040** | **0.5539** | **0.3747** | **0.2330** | **0.9042** | **0.9449** | **0.8651** |

**Effectiveness of the multi-view unified structure guidance** To verify the effectiveness of the unified structure guidance, we conduct experiments on all datasets. As shown in Table 3, "View-specific" indicates that different structure guidance is constructed for each view representation learning. "Unified" represents our proposed strategy, where a unified structure guidance is shared across multiple views through multi-step random walks on the constructed heterogeneous graph. "None" means that a unit matrix is used for warm start during the learning process. It can be observed that using the unit matrix as the initial structural target can prevent the collapse of representations. Moreover, utilizing view-specific structure guidance, DIVIDE (Lu et al., 2024) can capture the relationships between samples within each view. However, due to the independence between views, the structure in each view may not be consistent, potentially introducing structural noise that could harm clustering performance. Our proposed SMvCNet constructs a heterogeneous graph for multi-view data and employs a multi-step random walk strategy to capture the complementary information across views while preserving view consistency. This provides a unified structure guidance for multi-view learning, making it better suited for downstream clustering tasks.

Table 4: Ablation studies on two structure learning losses.

| $\mathcal{L}_{within}$ | $\mathcal{L}_{cross}$ | Scene15 | | | UCI-Digit | | | CUB | | | Prokaryotic | | | ALOI | | |
|---|---|---|---|---|---|---|---|---|---|---|---|---|---|---|---|---|
| | | ACC | NMI | ARI | ACC | NMI | ARI | ACC | NMI | ARI | ACC | NMI | ARI | ACC | NMI | ARI |
| ✓ | | 0.4520 | 0.4571 | 0.2808 | 0.9531 | 0.9073 | 0.8995 | 0.8233 | **0.8043** | **0.7121** | **0.5590** | **0.3927** | **0.2415** | 0.8683 | 0.9273 | 0.8227 |
| | ✓ | 0.4713 | **0.4888** | 0.3098 | **0.9589** | **0.9152** | **0.9108** | 0.7480 | 0.7261 | 0.6078 | 0.5053 | 0.2356 | 0.1436 | 0.8785 | 0.9141 | 0.8002 |
| ✓ | ✓ | **0.4808** | 0.4887 | **0.3109** | 0.9538 | 0.9102 | 0.9007 | **0.8317** | 0.7891 | 0.7040 | 0.5539 | 0.3747 | 0.2330 | **0.9042** | **0.9449** | **0.8651** |

**Effectiveness of the two structure learning losses** To demonstrate the effectiveness of the structure learning losses, we conducted an ablation study on the losses as shown in Table 4. Based on the results, we have the following findings: For datasets with large differences in view dimensions and relatively small sizes, such as CUB and Prokaryotic datasets, the representation learning is mainly achieved through the optimization of within-view loss. It is evident that the cross-view loss still exhibits significant inconsistency on these datasets, making it challenging to enforce multi-view structural consistency constraints. For datasets with smaller differences in view dimensions, such as UCI-Digit dataset, the optimization directions of within-view loss and cross-view loss are generally consistent. As a result, the impact of different losses on performance is not significant. Furthermore, on multi-view datasets containing a large number of samples and clusters, such as Scene15 and ALOI datasets, the two losses we used exhibit different effects. However, when both losses are combined, a more significant performance improvement is achieved. In summary, although the two structural losses demonstrate varying effects on different datasets, considering a balance between the two is more beneficial for multi-view representation learning and clustering tasks.

## 4.5 PARAMETER ANALYSIS (Q3)

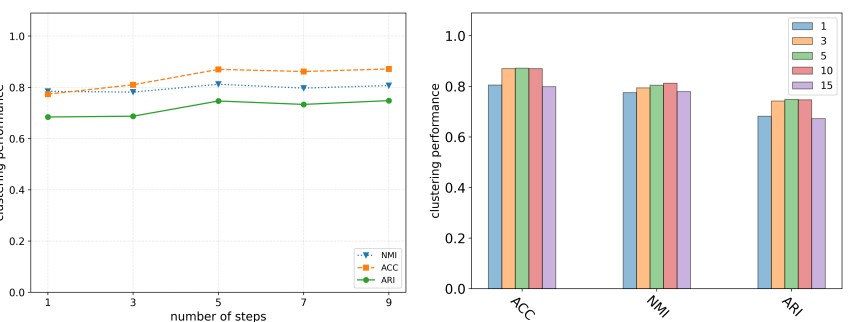

(a) The impact of steps of random walks.   (b) The impact of rounds of random walks.

Figure 3: Parameters study on CUB dataset.

We conducted experiments on CUB dataset to examine the effects of the number of steps $t$ and the number of rounds $n_{rw}$ of proposed multi-view random walks. As shown in Figure 3(a), as the

number of steps in the random walk increases, i.e., when structural information of different orders is introduced, the clustering performance first improves and then stabilizes. On the CUB dataset, the introduction of 5th-order structure information significantly enhances performance, indicating that compared to lower-order information (1st and 3rd orders), it contains more effective information that helps capture relationships among data samples. As the order increases further, performance tends to stabilize, suggesting that the additional higher-order sample structure contain less effective information and may even introduce some noise. Overall, introducing 5th-order ($t$=5) structure information is sufficient to multi-view clustering. Through observing Figure 3(b), we found that the number of random walk rounds follows a similar trend. In the experiments, we set $n_{\mathrm{rw}}$ to 5 to ensure applicability across all datasets.

## 4.6 STRUCTURE VISUALIZATION (Q4)

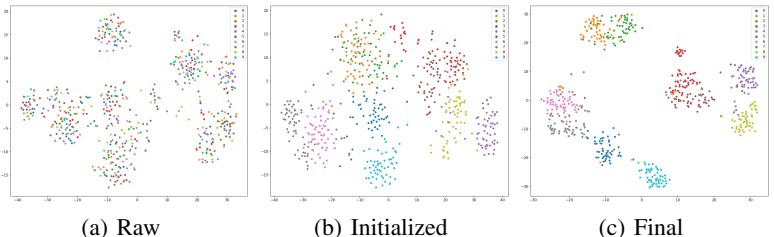

|       (a) Raw       |   (b) Initialized   |   (c) Final   |

Figure 4: Visualization of data structure on CUB.

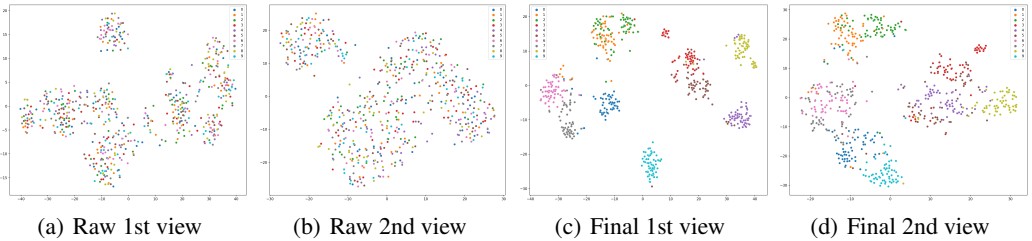

|  (a) Raw 1st view  |  (b) Raw 2nd view  |  (c) Final 1st view  |  (d) Final 2nd view  |

Figure 5: Visualization of view structure on CUB.

In this part, we conduct the visualization experiment to demonstrate the superiority of proposed SMvCNet intuitively. To be specific, we visualize the data structure and view structure of SMvCNet on CUB dataset via t-SNE algorithm (Van der Maaten & Hinton, 2008). From the results in Figures 4 and 5, we concluded that compared with the raw and initialized structures, the proposed SMvCNet better reveals the intrinsic clustering structure at both the data and view levels.

## 4.7 PRUNING EXPERIMENT (Q5)

To improve the effectiveness of the proposed SMvCNet and to lower the complexity of the designed random walks on constructed heterogeneous graph, we also explored a graph simplification strategy. This strategy entails reducing computational complexity by either pruning nodes or edges.

**Node Pruning**   For node pruning, we adopted a simple approach based on random sampling to filter nodes. We evaluated four clustering metrics (including PURITY) on different scales of datasets CUB and UCI-Digit, by performing heterogeneous random walks using 70%, 80%, 90%, and all nodes. As depicted in Figure 6, increasing the number of nodes allows for capturing more structure information, thereby enhancing performance. Additionally, the comparison between Figures 6(a) and 6(b) indicates that the node pruning strategy has a more noticeable impact on smaller datasets such as CUB, while its effect on the UCI-Digit dataset is relatively limited.

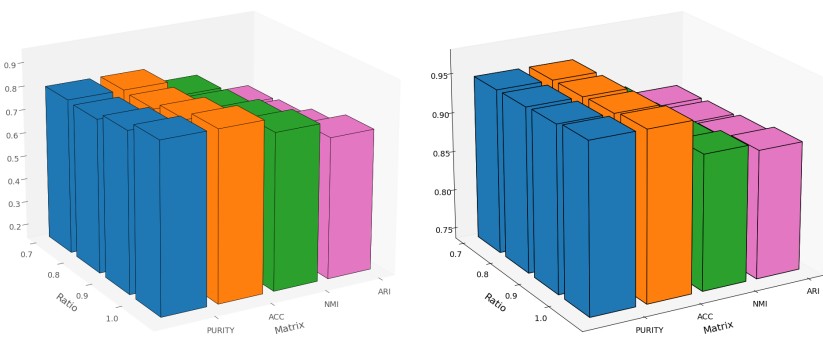

(a) The impact of nodes ratio on CUB.    (b) The impact of nodes ratio on UCI-Digit.

Figure 6: Experiments with node pruning.

**Edge Pruning**    For edge pruning, we utilized the $k$-nearest neighbor ($k$-nn) method to filter edges. We evaluated the clustering performance (NMI) of multi-view random walks on the CUB and UCI-Digits datasets by comparing constructions with 1-nn, 5-nn, 10-nn, and 20-nn, as well as the complete graph (all-nn). Figure 7 illustrates that as $k$ increases from 1 to 20, the NMI consistently rises. This suggests that incorporating more effective neighboring edges captures greater structural information. Notably, when comparing $k$-nn graph to the complete graph random walks on the CUB dataset (Figure 7(a)), the complete graph proves to be more beneficial for exploring the structure of all samples. Conversely, on the UCI-Digit dataset (Figure 7(b)), the effective $k$-nn graph is better at removing irrelevant edges, leading to improved clustering performance.

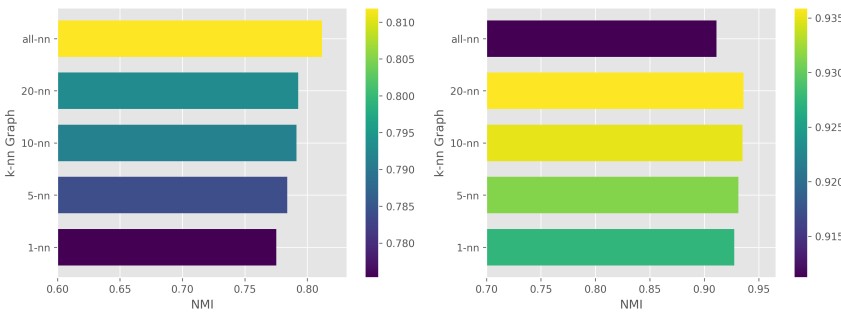

(a) The impact of $k$-nn graph on CUB.    (b) The impact of $k$-nn graph on UCI-Digit.

Figure 7: Experiments with edge pruning.

In practical scenarios, we can adopt different graph simplification strategies according to the unique characteristics of each dataset.

## 5    CONCLUSIONS

This paper proposes a structural multi-view clustering network via heterogeneous random walks. By constructing a heterogeneous graph and employing a multi-step random walk, we uncover high-order sample structures across various views while preserving view-specific structures. The network is guided by the unified structure derived from heterogeneous random walks, enhancing representation and improving clustering performance. Extensive experiments demonstrate the superiority of SMvCNet over existing models. In future work, we will further explore graph simplification strategies to improve efficiency and effectiveness of designed heterogeneous random walks.

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
