# OpenReview forum: "Structural Multi-view Clustering Network via Heterogeneous Random Walks"
_ICLR.cc/2025/Conference — ICLR 2025 Conference Withdrawn Submission_

### Official Review · Reviewer_B3GN · 2024-11-01

**Soundness:** 2
**Presentation:** 2
**Contribution:** 1
**Rating:** 3
**Confidence:** 4

**Summary:**

1.	For multi-view data, the authors construct a heterogeneous graph to capture similarity within the same view and the mapping relations among views. A random walk process is formulated among the sample nodes.
2.	With Siamese encoders for each view and cross-view projectors, the proposed model learns representations with proximity relations that align with the heterogeneous random walk process.
3.	Experiments conducted on five datasets show that the proposed SMvCNet outperforms a series of baselines. Ablation studies and parameter analysis are also given.

**Strengths:**

1.	The construction of heterogeneous graph from multi-view data is well-defined and theoretically sound.
2.	A unified random walk process is a good strategy for aligning common information among multiple views for clustering.
3.	The experiments section is well organized with 5 research questions.

**Weaknesses:**

1.	The formulation of heterogeneous random walk is, in essence, the average of proximity graph transition matrices from all views. Mixing the random walk process from multiple views has been studied in “Spectral Clustering and Transductive Learning with Multiple Views.” (Zhou et al., 2017)
2.	The methodology in this work bears substantial similarities to the DIVIDE method in “Decoupled Contrastive Multi-View Clustering with High-Order Random Walks”. (Lu et al., 2024) The random walk-based proximity measure, model architecture and loss functions in SMvCNet largely follow from DIVIDE. There is one noticeable difference, that the heterogeneous random walk is adopted instead of each view’s own random walk transition matrix. Hence, the novelty of this work should be clarified.
3.	The presentation of the SMvCNet clustering method should be more complete.  In Figure 2, “Momentum encoder” modules and “Moving average” procedures are never addressed in the text. The paper also does not explain how the final clustering results are acquired.
4.	Given the wide range of multi-view datasets, the experiment results could be strengthened by including more data. Especially, the number of views here is rather small, on average 2.8.

**Questions:**

1.	Given that SMvCNet adopts pairwise similarity matrices, multi-hop transition matrix and cross-view projections, the scalability might be affected. What is the computational complexity of this clustering method? Are there empirical results of clustering efficiency against the baseline methods? Is the edge pruning technique in 4.7 effective for improving efficiency?
2.	In the paper “Decoupled Contrastive Multi-View Clustering with High-Order Random Walks” (Lu et al., 2024) the ACC/NMI/ARI results on the complete Scene15 dataset are 0.491, 0.487, 0.316, respectively. However, the metrics reported here are much lower (0.431, 0.435, 0.259). ACCMVC’s results on the Prokaryotic dataset are also much lower than those reported in their original paper.
3.	More recent works on multi-view clustering could be compared and discussed. For example, “Cross-view Topology Based Consistent and Complementary Information for Deep Multi-view Clustering” (Dong et al., 2023) also reports competitive results on the UCI-Digit dataset.

---

### Official Review · Reviewer_cP79 · 2024-11-02

**Soundness:** 1
**Presentation:** 2
**Contribution:** 3
**Rating:** 3
**Confidence:** 5

**Summary:**

By constructing a heterogeneous graph that is consistent with sample nodes and view nodes, the authors propose a framework by utilizing the random walk strategy for learning view consensus and high-order structural information. The authors claim the proposed heterogeneous graph can obtain cross-view information so that to enhance the clustering performance. Experimental results achieved comparative performance on five datasets.

**Strengths:**

The idea of constructing a heterogeneous graph is interesting. The introduction of sample nodes among different views can be seem as a bridge for multi-view interaction, and random walk strategy can align well with the requirements for learning the heterogeneous graph.

**Weaknesses:**

1. [Contribution] This work is an incremental contribution compared to DIVIDE (2024' CVPR), although the idea of the heterogeneous graph is somehow novel. The framework and training strategy of the proposed method are the same as those of DIVIDE.
2. [Soundness] The advantage and effectiveness of the proposed heterogeneous graph (which is the main contribution of this paper) are not clearly presented and demonstrated. The authors only show the effectiveness in an ablation study (Table 3), but this is not sufficient for showing its effectiveness. I would suggest the author to deeply analyze and discuss why it works and how it works.
3. [Motivation] The motivation is not clearly presented, e.g., why they propose the heterogeneous graph, what's the difference between introducing node samples for interaction among views and directly learning among views?
4. [Presentation] The organization of this paper is not good, especially the Methods section. For example, the matrix $M$ is defined two times,"the transition matrix M of v-th view is obtained by normalizing the adjacency matrix A in a row-wise manner" and Eq.(10). The author should formally define it when it first appear. $T$ is not clearly defined, what's it value range?
5. [Presentation] Contribution is not clear. The authors referred to many designs from DIVIDE, what's the main novel and main contribution of this paper?

**Questions:**

Please check weakness

---

### Official Review · Reviewer_YY17 · 2024-11-04

**Soundness:** 3
**Presentation:** 3
**Contribution:** 2
**Rating:** 6
**Confidence:** 3

**Summary:**

This work studies multi-view clustering which is an important topic. The authors propose a structural multi-view clustering network via heterogeneous random walks. The extensive ablation studies may be a valuable contribution to the community.

**Strengths:**

- The investigated problem holds significant value in the domain of multi-view clustering and is also of great interest to the research community.
- This paper is well written and easy to read and understand.
- Experiments are done on clustering tasks using different methods, ensuring a comprehensive evaluation.

**Weaknesses:**

- The time and space complexities for Algorithm 1 and the other methods should be given.

- Is there any theoretical support for the proposed algorithm expected performance improvement?

- How about the convergence of the algorithm? Is there any convergence proof or theoretical guarantee?

- I would appreciate a broader discussion on why the proposed method performs better than the others, for well understanding the connection and difference among existing methods.

**Questions:**

Please see Weaknesses.

---

### Note · Authors · 2024-11-23

I have read and agree with the venue's withdrawal policy on behalf of myself and my co-authors.